# State-Space Modeling and Analysis for an Inverter-Based Intelligent Microgrid under Parametric Uncertainty

**Yiwei Feng \* and Zong Ma** 🆔

School of Electrical Engineering and Information Engineering, Lanzhou University of Technology, Lanzhou 730000, China
* Correspondence: ywfeng@yeah.net

**Abstract:** In this paper, a multivariable linear integral feedback regulation controller for a microgrid was proposed. Considering that the nominal structure model of the inverter could not effectively and in a timely manner deal with the impact of filter parameter uncertainty, there were changes in output power quality among different generation environments. To solve the constraints imposed by uncertain factors on the system, we formulated the following scheme. First, based on the analysis of the asymptotic stability and power characteristics of the nominal model, we added the microgrid filter parameter uncertainty to this model. Secondly, under the action of the bounded range, the performance characteristics of the optimal cost were analyzed, adjusted, and optimized. The controller adjusted parameters to ensure the stable operation of the microgrid system, and to achieve the voltage stability regulation and output power balance. Finally, we built a test system to verify the feasibility and effectiveness of the proposed linear integral controller in MATLAB/Simulink.

**Keywords:** linear integral regulation controller; nominal model; parameter uncertainty; bounded range analysis; output power balance

## 1. Introduction

In recent years, with the rise of renewable energy Systems (RESs) and distributed generation (DG), advances in power converters and digital control technology have made it particularly important to integrate RESs into modern power systems. It is feasible to use distributed generation to provide a continuous power supply for key loads after power failure in large power grids and main power grids [1,2]. In order to reduce the pollution and waste of the power grid, the power grid connected to the power system through an inverter or converter should have a high output power factor and a low output current distortion [1]. At first, an inverter or a converter with a high power density was designed to connect the DG units of RESs to the large public power grid [3]. However, because of the excessive switching loss and the upper limit of the switching equipment, the output of power quality was reduced [4]; we used a compact high-order filter to make up for this disadvantage [5]. Compared with a traditional inverter connection mode, an inverter with a filter has the following advantages: a small inductance size, better attenuation of high-frequency harmonics, and lower power and current ripple. Similarly, the use of filters brought new challenges to the control methods and power grid structure [6]. For example, because of the uncertainty and interference problems such as frequency voltage, control loop delay, and structural nonlinearity, standard sinusoidal current had to be realized under the conditions of grid voltage harmonics and non–ideal circuit components to avoid resonance. This ensured that the system's stable and robust performance was ascertained [7]. At present, although there is no comprehensive method to solve these emerging challenges, various approaches have been proposed to solve them in a collaborative way.

Smart microgrids combine communication technology with control technology to achieve smarter, more efficient and robust large power networks. The first step in the

design process of a smart microgrid is to establish a simulation nominal model under ideal conditions [6]. Using inherent stability analysis methods, such as eigenvalue analysis and singular value analysis, a controller is designed to verify the accuracy and robustness of the microgrid system model. However, the traditional synchronous reference frame phase-locked loop (SRF-PLL) method could lead to frequency fluctuations and degrade the system performance. To improve the power quality and recovery ability of the power injected into the grid in a distortion environment, frequency, voltage, and current deviation problems are studied as a key part of power system research. The authors [8] used a moving average filter (MAF) to eliminate the frequency fluctuation of the PLL, and to compensate for the harmonic distortion caused by the power grid voltage distortion, but the control effect on the current was small and the algorithm of the MAF was relatively complex. The authors [9] proposed a method for the stability domain in island and grid connection modes to optimize and calculate the maximum limit value of the stability domain and to achieve the robust stability of the system. However, this method was only suitable for calculating two known uncertain parameters. The work of [10] studied the multi-loop control scheme, which combined a repetitive controller (RC), resonant controller, and a grid voltage feed-forward controller. This work achieved the seamless conversion between the grid-connected and island modes and made the current stable, eliminating the harmonic distortion of voltage and frequency change. However, the requirement on the device level was high; it was limited to small electronic devices and practical projects. For current harmonics in [11], a design RC could compensate for the total harmonic distortion (THD) and improve the dynamic response. However, to precisely track the fundamental frequency and harmonics was very difficult, which limited the bandwidth of the controller and existing harmonics in order to eliminate the selected harmonic components. The authors of [12] used the harmony search (HS) algorithm based on H-infinity to optimize frequency and voltage, and to make the system more robust and stable. However, the algorithm does not consider the impact of time delay on the system failure or load change. The work in [13] can compensate for the total harmonic distortion (THD), but due to the high gain at the fundamental frequency and harmonic frequency, the zero steady-state tracking error was linearly eliminated. In [14] the authors designed harmonic compensators and configured them in a series with the tracking regulator of single-phase grid-connected inverters. This effectively attenuated the voltage distortion and accurately synchronized the tracking. Thus, the high gain at the fundamental frequency and harmonic frequency, as well as the zero steady-state tracking error were eliminated. However, the sensitivity of the disturbance to power quality and the uncertainties of the filter parameters were not considered. The authors of [15], proposed an adaptive control scheme that used a highly accurate track with existing inherent resonance and voltage harmonic distortion and gave the power grid a better, more robust performance. Smart microgrid systems need to consider more dynamic, complex external interference and uncertain input factors to be able to respond to the needs of practical engineering. Therefore, the improvement of power quality under nominal conditions has limitations. For example, the load frequency quadratic control method in [16] that is based on an unknown stochastic input observer and linear quadratic regulator (LQR), not only effectively solved the uncertainties, such as load variation and measurement noise, but also produced better robustness. Although the research on filter-based powers system is convenient for solving a lot of problems, the filter is attached with complex conjugate poles and resonance damping requirements, which makes the power system very sensitive to parameter uncertainty and the design of its control strategy more complex. In [17], the authors proposed a virtual inductance method to improve the stability of the microgrid under a constant power load. However, whether the control strategy can be extended to the AC/DC hybrid microgrid was not considered. The work of [18] considered distributed quadratic consistency and fault-tolerant control to compensate for the frequency and voltage errors and to achieve accurate power distribution. However, the influence of the involved load change or fault was ignored. The author [19] designed an efficient and reliable intelligent control strategy for the secondary reconfigurable inverter for the off grid microgrid and diagnosed the

fault. In [2], the authors proposed a robust nested loop control scheme, with different load and filter parameter uncertainties that enabled two types of modes to operate stably; however, this required an accurate mathematical model. In [20], the authors designed a centralized robust controller through a suboptimal solution of a convex optimization problem to resolve the influence of parameter uncertainty and unknown disturbances. However, there was no bounded analysis of parameters, and the time delay was not considered. In [21], a new dynamic sliding mode control (DSMC) solved the shortcomings of traditional control state references and sliding coefficient calculation. This compensated for the impact of time delay and uncertainty in the system, so the system had robustness and stability. However, the authors did not consider the capacitor voltage variation of the load itself. The authors of [22], based on droop control, designed a sliding mode control method to overcome the difference caused by feedback line impedance, reduce the error of voltage and reactive power, and improve the accuracy of data adjustment. In [23], the authors proposed that the stabilization algorithm was bound to deal with a class of boundless uncertainties, but the algorithm was too simple and not applicable to various types of uncertainties. In [24,25], robust state feedback control and steady-state robust state estimation for uncertain linear systems were designed by the authors to refine the types of uncertainties and to achieve better cost estimates, but the non-linear systems were not considered. The work of [26], a grid connected inverter control technology based on dq transformation, was introduced, which reduced the cost and the change of power harmonics, ensured the power quality, and made the power output stable; but it did not consider the impacts of the LCL filter and inverter structures on the system oscillation and output distortion. In [27], the authors designed an event triggered robust controller to solve the problem of frequency instability caused by uncertainty, and used system interference attenuation to reduce the communication burden. However, this method needed an accurate energy storage device model and did not consider the influence of the transmission delay. The work of [28], the distributed self-triggering secondary control, was proposed to solve the impact of communication interference and realize voltage recovery and reactive power sharing. The work of [29] achieved the bounded stability of frequency and voltage after an uncertain load change. However, it lacked the impact of the sensitivity of the hardware—in—the loop (HIL) on the boundary. In [30], the authors proposed online primary regulation under island to reduce power loss, optimize operation characteristics, and improve system reliability. However, the parameter sensitivity of the HIL was not considered. In [31], the authors investigated robust optimal control for a class of nonlinear quadratic systems with norm-bounded parameter uncertainties and disturbances. The work solved and suppressed the influence of disturbances and achieved system stability. However, this method had great limitations as it only realized the system in the local stability of the equilibrium point it did not apply to the actual engineering. Therefore, in [32], a newly defined LMI structure was introduced, which studied the design of the controller and observer for uncertain positive discrete systems, so that the ideal model could respond to actual engineering requirements and achieve secondary stability control.

As mentioned above, the intermittency of distributed energy affects the output of the intelligent microgrid inverter, leading to changes in power quality. The inverter with filter has a better control output effect than the traditional inverter and can respond to the demand of the power system in real time. However, because the resonance damping and conjugate poles of the filter are sensitive to parameters, parameter uncertainty will interfere with the stable operation of the microgrid system, resulting in voltage distortion and power imbalance. To better meet the needs of practical projects, the main contribution of this paper is to design a moving average phase-locked loop based on the nominal model to replace the traditional phase-locked loop to limit frequency fluctuations. Secondly, when the filter parameters change, based on the guaranteed cost control theory and method, this kind of uncertainty problem is analyzed in a bounded range to achieve a low-cost filter. Finally, the linear integral quadratic regulation (LQR) state feedback controller is used to optimize the influence of uncertain parameters on the system model, suppress the voltage

harmonic distortion, ensure that the system is in a certain progressive stable state, and ensure the power output balance.

The paper is arranged as follows: Section 2 describes the problem and analyzes the model and the bounded range of parameter uncertainty according to assumptions. Section 3 describes a linear integral controller that is designed to analyze the stability and verify the hypothesis; Section 4 addresses the optimization and adjustment system parameters that verify system reliability. Section 5 summarizes the conclusions and prospects for future work.

## 2. Problem Description and Model Analysis

This section discusses the system operation state under parameter uncertainty based on the nominal model. First, the limitations of the nominal model are discussed. Secondly, state space modeling is carried out. Finally, combined with relevant research, the system is assumed to be stable and the bounded uncertainty range is provided.

### 2.1. A Configuration of PLL

The peer-to-peer control strategy in island mode in the intelligent micro network is necessary as shown in Figure 1, which means that micro power supplies in MG have the same control state. First, each micro power supply performs local control according to the voltage and frequency of the access point. Plug and play are realized and automatized adjustment of voltage and frequency takes place, eliminating communication, improving the reliability of MG, and reducing costs.

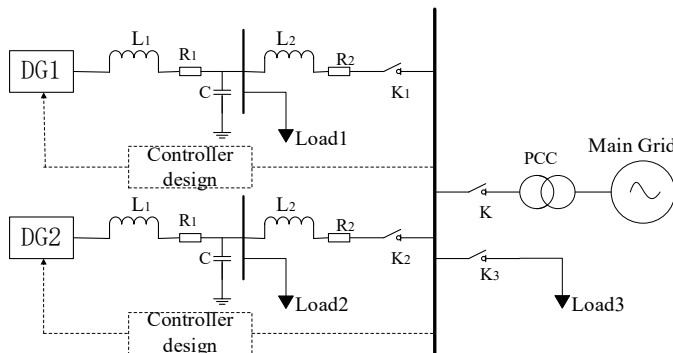

**Figure 1.** Microgrid peer-to-peer control.

However, this control strategy has certain limitations that prevent it from eliminating the harmonic distortion and frequency change in voltage, current, and power characteristics of the distributed generator and power grid to some extent and ensuring the effectiveness and feasibility of model verification.

An accurate estimation of power grid frequency was a key step, so moving the average filter (MAF) is designed to eliminate the frequency fluctuation in phase-locked loop in the synchronous reference frame as follow Figure 2. The MAF transfer function is:

$$G_{MAF}(z) = \frac{1}{N} \bullet \frac{1 - z^{-N}}{1 - z^{-1}} \tag{1}$$

where $N = T_w/T_s$ is the number of samples within the window length of MAF, and $T_w$ is the window length of MAF.

As the uncertainty of three-phase voltage is mostly odd harmonics, and even harmonics are generated in SRF, half of the base period was selected as the window length. The number of samples is $N = T/2T_s$, where $t$ is the fundamental frequency period of grid voltage [15]. The state space equation of the integral control term is as follows:

$$z_0(k + 1) = A_p z_0(k) + B_p \varepsilon(k) \tag{2}$$

where $A_p = I_{2\times2}$, $B_p = T_s \times I_{2\times2}$, $z_0(k+1)$ is the state vector of the integral control term, $\varepsilon(k) = \begin{bmatrix} i_{2q}^*(k) \\ i_{2d}^*(k) \end{bmatrix} - C_d x_s(k)$ is the current error vector, $i_2^*(k)$ is the reference vector.

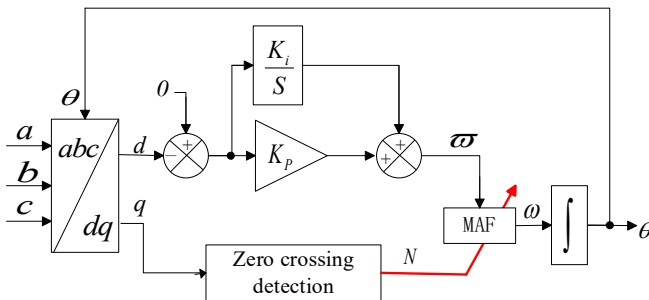

**Figure 2.** Configuration of the MAF-PLL scheme.

The improved nominal model was analyzed. First, a unit was analyzed that as shown in Figure 3, obtain the stable changes of frequency and voltage of the power grid unit.

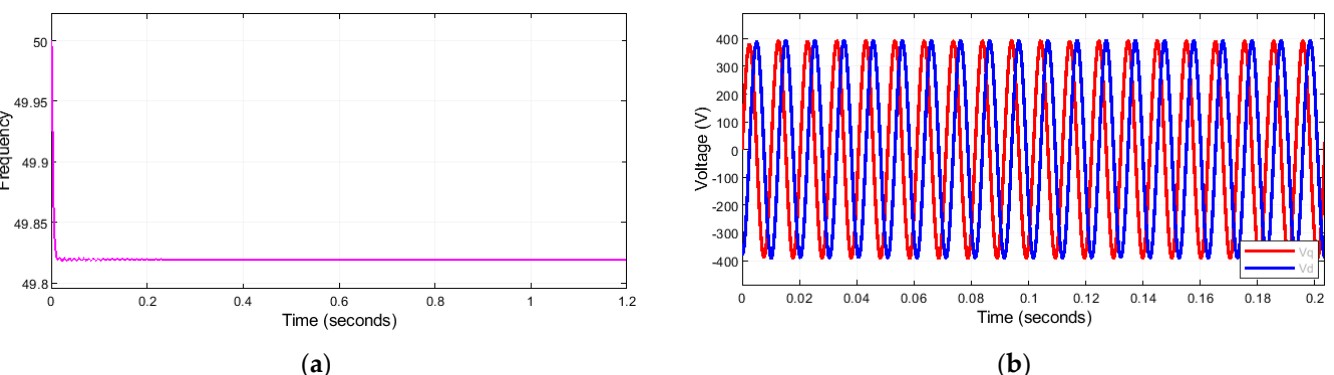

**Figure 3.** A separate inverter unit: (**a**) Frequency; (**b**) Voltage.

The parameters of connecting between the DG unit and the main grid are as follows: According to Table 1, two DG devices were used in parallel to obtain stable voltage and power as shown in Figures 4 and 5.

**Table 1.** System parameters.

| Description | Symbol | Value |
| --- | --- | --- |
| Converter side inductance of filter | $L_1$ | 1.83 mH |
| Converter side resistance of filter | $R_1$ | 0.52 mΩ |
| Grid side inductance of filter | $L_2$ | 1.75 mH |
| Grid side resistance of filter | $R_2$ | 12 mΩ |
| Damping resistance of filter | $R_f$ | 0.6 Ω |
| Capacitance of filter | $C_f$ | 270 μF |

### 2.2. State Space Model

Compared with the application in practical control engineering, the uncertainty of system inductance and capacitance inductance (LCL) were studied. It was proposed that adding components with impedance or parameters to the filter instead of uncertain factors would change the characteristics of the nominal system.

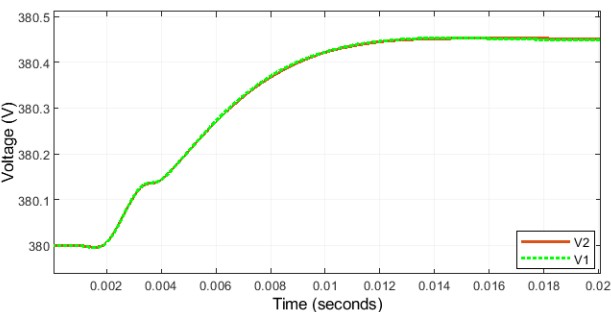

**Figure 4.** Voltage changes after the parallel connection of inverter $V_{DGi} = V1$ ($i = 1, 2$).

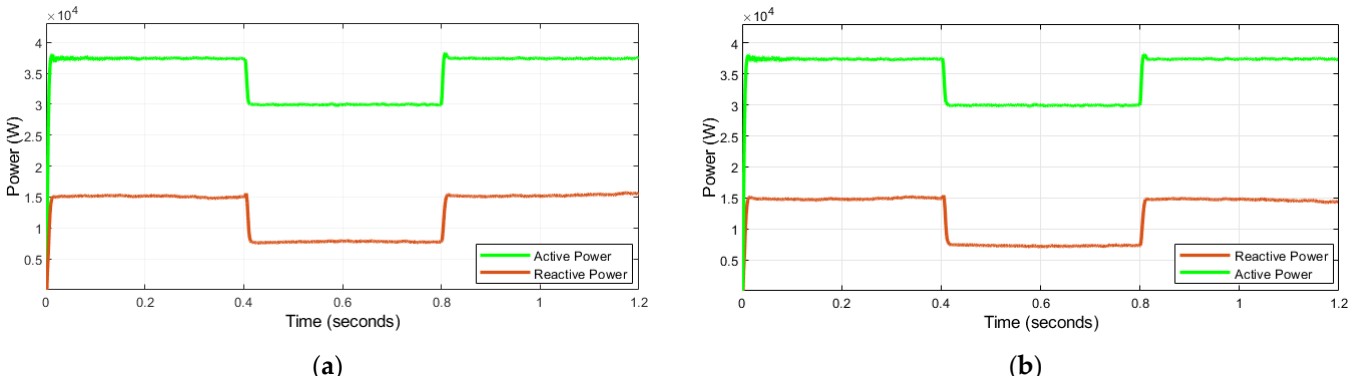

(**a**)

(**b**)

**Figure 5.** Active and reactive power under nominal model: (**a**) DG1; (**b**) DG2.

For the accurate modeling and performance analysis of an uncertain intelligent microgrid system. Combined with Figure 6 and the specific filter uncertain parameters were given to establish the state-space model of the inverter in the synchronous coordinate system, as shown below:

$$\frac{di_{1,d}}{dt} = \frac{1}{L_1}\left(v_d - \left(R_1 + R_f\right)i_{1,d} + +\omega L_1 i_{1,q} - v_{f,d}\right) \tag{3}$$

$$\frac{di_{1,q}}{dt} = \frac{1}{L_1}\left(v_q - \left(R_1 + R_f\right)i_{1,q} + \omega L_1 i_{1,d} - v_{f,q}\right) \tag{4}$$

$$\frac{di_{2,d}}{dt} = \frac{1}{L_2}\left(v_f + R_f i_{1,d} - \left(R_2 + R_f\right)i_{2,q} + \omega L_{tot} i_{2,q}\right) \tag{5}$$

$$\frac{di_{2,q}}{dt} = \frac{1}{L_2}\left(v_f + R_f i_{1,q} - \left(R_2 + R_f\right)i_{2,q} - \omega L_{tot} i_{2,d}\right) \tag{6}$$

$$\frac{dv_{f,d}}{dt} = \frac{1}{C_f}\left(i_{1,d} - i_{2,d} + \omega C_f v_{f,q}\right) \tag{7}$$

$$\frac{dv_{f,q}}{dt} = \frac{1}{C_f}\left(i_{1,q} - i_{2,q} - \omega C_f v_{f,d}\right) \tag{8}$$

where $L_1$, $R_1$, $L_2$, $R_2$ are the impedance, $C_f$ and $R_f$ are the capacitance and impedance, $v_{f,q}$, $v_{f,q}$ are the capacitance $v_c$ shunt voltage at both ends, $v_{g,d}$, $v_{g,q}$ are the grid voltage $v_g$ shunt voltage at both ends, between the microgrid side and the grid side branch current $i_{1,d}$, $i_{1,q}$, $i_{2,d}$, $i_{2,q}$, $\omega$ is the power system frequency.

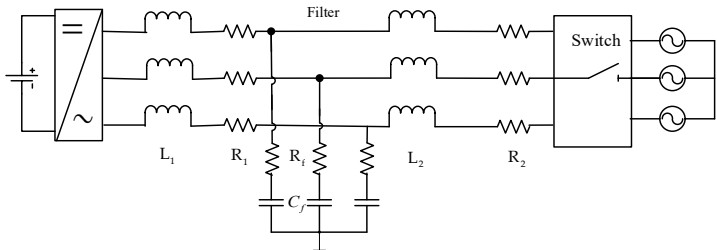

**Figure 6.** Configuration filter of a grid connected inverter with an inductor capacitor inductor.

At this time, the continuous time representation of the system can be re represented by the state-space model of the following formula:

$$\begin{cases} \dot{x}(t) = A^*(t)x(t) + B_1^*(t)U_{dq}(t) + B_2^*(t)e_{gdq}(t) \\ y(t) = C(t)x(t) \end{cases} \tag{9}$$

where $x_i = \begin{bmatrix} v_{fd} \ v_{fq} \ i_{1d} \ i_{1q} i_{2d} \ i_{2q} \end{bmatrix}^T$, $A^* = A_{dq} + \Delta A$, $B_1^* = B_{1dq} + \Delta B_1$,

$$B_2^* = B_{2dq} + \Delta B_2$$

$$B_{1dq} = \begin{bmatrix} 0 & 0 & 1/L_1 & 0 & 0 & 0 \\ 0 & 0 & 0 & 1/L_1 & 0 & 0 \end{bmatrix}^T, \quad B_{2dq} = \begin{bmatrix} 0 & 0 & 0 & 0 & 1/L_2 & 0 \\ 0 & 0 & 0 & 0 & 0 & 1/L_2 \end{bmatrix}^T,$$

$$A_{dq} = \begin{bmatrix} 0 & \omega_c & 1/C_f & 0 & -1/C_f & 0 \\ -\omega_c & 0 & 0 & 1/C_f & 0 & -1/C_f \\ -1/L_1 & 0 & -(R_1+R_f)/L_1 & \omega_c & R_f/L_1 & 0 \\ 0 & -1/L_1 & -\omega_c & -(R_1+R_f)/L_1 & 0 & R_f/L_1 \\ 1/L_2 & 0 & R_f/L_2 & 0 & -(R_2+R_f)/L_2 & \omega_c \\ 0 & 1/L_2 & 0 & R_f/L_2 & -\omega_c & -(R_2+R_f)/L_2 \end{bmatrix}$$

The grid side current was the measured state, and the inverter side current and capacitor voltage were the unmeasured parts of the state vector. To facilitate the use of data acquisition equipment and PWM signal acquisition in the simulation experiment, the bounded range analysis of parameter uncertainty was conducted.

*2.3. Bounded Analysis*

In order to achieve dynamic stability and robustness for uncertainty, a class of systems with norm bounded uncertainties were introduced into the H∞ control theory to ensure that the closed-loop system was asymptotically stable at the specified H∞ disturbance attenuation level [20]. But it is difficult to achieve a stable and uncertain optimal cost. So, for the non-zero term uncertainty, we assume that there is a positive scalar constant and a certain value limit. Therefore, we define the following:

$$S = \left\{ A^*(t), \ B_1^*(t), \ B_2^*(t), \ C(t) \right\} \tag{10}$$

Then, if $\hat{\Theta}(t) \in S$, we have that $\hat{\Theta}(t) = \hat{\Theta} + \Delta\hat{\Theta}(t)$, $\hat{\Theta} = \frac{(\theta_{ijmax}) + (\theta_{ijmin})}{2}$ is a time invariant mean matrix, and $\theta_{ij}$ is a matrix $\hat{\Theta}$ The element with the largest singular value in, $\Delta\hat{\Theta}(t)$ is a real valued uncertain matrix function.

**Assumption 1.** *When the number of uncertain resistive components or uncertain parameters are added to the LCL, the islanding system is stable. There is a reasonable stability margin $(\| (I + E_0 T)^{-1} \|_\infty \leq 1)$ established, then the sensitivity of nominal and uncertain perturbation is not much different, and the uncertainty is expressed in the following structured bounded form.*

$$\Delta\hat{\Theta}(t) = \begin{bmatrix} \Delta A & \Delta B_1 \end{bmatrix} = EM \begin{bmatrix} F_a & F_b \end{bmatrix} \tag{11}$$

*where: E, $F_a$, $F_b$ are units of a known real constant, and M is an uncertain perturbation matrix that can be measured.*

Therefore, the system model changes to the following:

$$\dot{x}(t) = (A + BK + EM(F_a + F_b K))x(t) \tag{12}$$

The cost functions related to uncertain systems are as follows:

$$J = \int_0^\infty \left( x^T(t)Qx(t) + u^T(t)Ru(t) \right) dt \tag{13}$$

$Q = Q^T$ is a positive semi-definite symmetric matrix, $R = R^T$ is a positive definite symmetric matrix and is controllable for (A, B).

**Lemma 1 [2].** *For any $\epsilon > 0$, given the matrices Y, C and D with appropriate dimensions, and Y is a symmetric matrix, then $Y + CDM + DTMTCT < 0$. For any M, the following equation is satisfied.*

$$symm(x^T PYMCx) \le \epsilon x^T PYY^T Px + \epsilon^{-1} x^T C^T Cx \quad \forall x \in \mathbb{R}^n \tag{14}$$

**Lemma 2 [31].** *Considering the system uncertainty (without disturbance), given an admissible set D and cost function (13), it is assumed that there are some positive $\epsilon_1$, $\epsilon_2$. There is an invariant set if and only if $\epsilon > 0$, $\epsilon \subset \mathbb{R}^n$, $\epsilon \supset \mathcal{D}$, a symmetric positive definite matrix P and a matrix K. We then set the following formula:*

$$\begin{aligned} x^T\{Q + K^T RK + symm(P[A - BK] + [A - BK]^T P)\}x + \epsilon_1 x^T PDD^T Px + \\ \epsilon_1^{-1} x^T (E_1 - E_2 K)^T (E_1 - E_2 K)x + \epsilon_2 x^T P[I_n \otimes (x^T D)][I_n \otimes (D^T x)]Px < 0 \end{aligned} \tag{15}$$

**Definition 1.** *If the system has no other interference, the designed state-feedback controller is: $u(t) = Kx(t)$ to keep the uncertain system stable. Given the cost function (14), an admissible set $\mathcal{D}$ and a positive definite cost matrix $P > 0$, the following satisfied:*

$$x^T \left[ Q + K^T RK \right] + 2x^T [A + BK + EM(F_a + F_b K)]x < 0 \tag{16}$$

For all admissible uncertainties $\Delta\hat{\Theta}(t)$ in system (9), if the following conditions are satisfied, the feedback controller is a quadratic guaranteed cost controller with a related cost matrix.

The relationship between the above quadratic stability definition and quadratic stabilizability definition is extended to ensure performance.

**Theorem 1.** *Considering the system stability under the assumption, the control law is a quadratic guaranteed cost control with cost matrix $P > 0$. Then the closed-loop uncertain system (12) is quadratic stable at any time under constraints $M^T(t)M(t) \le I$. In addition, for all allowable uncertainties $\Delta\hat{\Theta}(t)$, the corresponding value of the cost function (13) satisfies the following bound:*

$$J \le x_0^T Px_0 \tag{17}$$

**Proof.** The uncertain system performance index (13) satisfies the boundary (17) in the initial state, and it is proved by Lemmas 1 and 2. Under the constraints $M^T(t)M(t) \le I$, Lyapunov function is considered in combination with the above two lemmas:

$$V(x) = x^T(t)Px(t)$$

Then:

$$\dot{V}(x) = x^{\mathrm{T}}(t)(A^{\mathrm{T}}P + PA)x(t)$$
$$= x^{\mathrm{T}}(t)([A + BK + EM(F_a + F_bK)]^{\mathrm{T}}P + P[A + BK + EM(F_a + F_bK)])x(t)$$
$$\leq x^{\mathrm{T}}\{symm(P[A + BK] + [A + BK]^{\mathrm{T}}P) + EM(F_a + F_bK)\}x + \epsilon_1 x^{\mathrm{T}}PDD^{\mathrm{T}}Px +$$
$$\epsilon_1{}^{-1}x^{\mathrm{T}}(E_1 - E_2K)^{\mathrm{T}}(E_1 - E_2K)x + \epsilon_2 x^{\mathrm{T}}P[I_n \otimes (x^{\mathrm{T}}D)][I_n \otimes (D^{\mathrm{T}}x)]Px$$
$$\leq x^{\mathrm{T}}\{P[A + BK] + [A + BK]^{\mathrm{T}}P + EM(F_a + F_bK)\}x$$
$$< -x^{\mathrm{T}}(t)(Q + K^{\mathrm{T}}RK)x(t)$$

It can be seen that the Lyapunov derivative is negative definite, then $(Q + K^{\mathrm{T}}RK) = -I$. As $Q$ and $R$ are positive definite real symmetric matrices, there is a solution $P$ of positive definite real symmetric matrix, so that the uncertain system (12) tends to be asymptotically stable.

It has been proved that the system is asymptotically stable. Then, when $x(\infty) = 0$, there are:

$$J = \int_0^{\infty}\left(x^{\mathrm{T}}(t)Qx(t) + u^{\mathrm{T}}(t)Ru(t)\right)dt \leq V(x(0)) - V(x(\infty)) = V(x(0)) = x_0{}^{\mathrm{T}}Px_0 \quad (18)$$

From the above conditions, $\Delta\hat{\Theta}(t)$, the lower and upper bounds can be displayed here as follows:

$$-h\hat{\Theta}(t) \leq EM\begin{bmatrix} F_a & F_b \end{bmatrix} \leq h\hat{\Theta}(t), \ h\hat{\Theta}(t) = \frac{(\theta_{ijmax}) - (\theta_{ijmin})}{2} \quad (19)$$

□

**Remark 1.** *(11) and constraint condition $M^{\mathrm{T}}(t)M(t) \leq I$ are admissible conditions, which are widely used to represent the parameter uncertainty in the control system. When other disturbances are not considered, the derivative of the quadratic Lyapunov function along the system trajectory is in the form of $V(x) = x^{\mathrm{T}}(t)Px(t)$ to prove and infer Lemma 2.*

**Remark 2.** *In uncertain structure matrices $E$, $F_a$, $F_b$ and $M$, according to the restricted condition of Lebesque measurable matrix value, the function conforms to the above operation within a certain bounded range.*

## 3. Controller Design

For uncertain systems, the Riccati equation method for constructing the optimal quadratic guaranteed cost controller under the Lyapunov function is given. Symmetric matrix $P$ is a stable Riccati differential equation solution of $A^{\mathrm{T}}P + PA + PMP + Q = 0$.

**Lemma 3.** *For any $\epsilon > 0$ and any $M$ satisfying Equation (16), then combined with Lemma 1, there are $x \in \mathbb{R}^n$, all of which have the following:*

$$0 \leq \epsilon x^{\mathrm{T}}PYY^{\mathrm{T}}Px + \epsilon^{-1}x^{\mathrm{T}}C^{\mathrm{T}}Cx - symm\left(x^{\mathrm{T}}PYMCx\right) \quad (20)$$

**Theorem 2.** *Given $Q \in \mathbb{R}^{n \times n}$ and $R \in \mathbb{R}^{n \times n}$, the symmetric matrix satisfies the Riccati equation and has a constant $\epsilon > 0$, in combination with references [5,6], the Riccati equation is as follows:*

$$\left(A + B(\epsilon R + F_b{}^{\mathrm{T}}F_b)^{-1}F_b{}^{\mathrm{T}}F_a\right)^{\mathrm{T}}P + P\left(A + B(\epsilon R + F_b{}^{\mathrm{T}}F_b)^{-1}F_b{}^{\mathrm{T}}F_a\right) + \epsilon PEE^{\mathrm{T}}P -$$
$$\epsilon PB(\epsilon R + F_b{}^{\mathrm{T}}F_b)^{-1}B^{\mathrm{T}}P + \frac{1}{\epsilon}F_a{}^{\mathrm{T}}\left(I - F_b(\epsilon R + F_b{}^{\mathrm{T}}F_b)^{-1}F_b{}^{\mathrm{T}}\right)F_a + Q = 0 \quad (21)$$

*With positive definite symmetric solution $P$, the uncertain system (10) is quadratic stable. Then, the appropriate stable control rate of the state feedback system is given as:*

$K = -\left(\epsilon R + F_b{}^\mathrm{T}F_b\right)^{-1}\left(\epsilon B^\mathrm{T}P - F_b{}^\mathrm{T}F_a\right) = -W^{-1}V$, *which meets the assumed design criteria and makes the system control stable.*

**Proof.** Combined with Lemma 1, for any $\epsilon > 0$ and any m satisfy Equation (16), there are $x \in \mathbb{R}^n$, all of which have the following:

$$0 \le \epsilon x^\mathrm{T}PYY^\mathrm{T}Px + \epsilon^{-1}x^\mathrm{T}C^\mathrm{T}Cx - symm\left(x^\mathrm{T}PYMCx\right)$$

For the Riccati equation, define $\overline{A} = A - BK$, $\widetilde{R}_2 = \epsilon R + F_b{}^\mathrm{T}F_b$, $\overline{E} = F_a + F_bK$, $\overline{Q} = F_a + K^\mathrm{T}RK$, $\hat{P} = \epsilon P$ and when $\widetilde{P} = (\epsilon/\epsilon_1)P = (1/\epsilon_1)\hat{P} > P$, any constant $\epsilon_1 < \frac{1}{n\overline{\sigma}[P(t)]} \in (0, \epsilon)$, According to the above expression, we have the following equation:

$$
\begin{aligned}
\overline{A}^\mathrm{T}P + P\overline{A} &+ \epsilon PEE^\mathrm{T}P + \epsilon PB\widetilde{R}_2{}^{-1}B^\mathrm{T}P + \tfrac{1}{\epsilon}F_a{}^\mathrm{T}\left(I - F_b\widetilde{R}_2{}^{-1}F_b{}^\mathrm{T}\right)F_a + Q \\
&= \overline{A}^\mathrm{T}P + P\overline{A} + \epsilon PEE^\mathrm{T}P + \epsilon PB\widetilde{R}_2{}^{-1}B^\mathrm{T}P + \tfrac{1}{\epsilon}\overline{E}^{-1}\overline{E} + \overline{Q} \\
&= \overline{A}^\mathrm{T}\hat{P} + \hat{P}\overline{A} + \hat{P}EE^\mathrm{T}\hat{P} + PB\widetilde{R}_2{}^{-1}B^\mathrm{T}P + \overline{E}^{-1}\overline{E} + \epsilon\overline{Q} \\
&= \overline{A}^\mathrm{T}\hat{P} + \hat{P}\overline{A} + \hat{P}EE^\mathrm{T}\hat{P} + PB\widetilde{R}_2{}^{-1}B^\mathrm{T}P + \overline{E}^{-1}\overline{E} + \epsilon\overline{Q} \\
&< \overline{A}^\mathrm{T}\hat{P} + \hat{P}\overline{A} + \hat{P}EE^\mathrm{T}\hat{P} + \overline{E}^{-1}\overline{E} + \epsilon_1\overline{Q} \\
&< \overline{A}^\mathrm{T}\widetilde{P} + \widetilde{P}\overline{A} + \epsilon_1\widetilde{P}EE^\mathrm{T}\widetilde{P} + \tfrac{1}{\epsilon_1}\overline{E}^{-1}\overline{E} + \overline{Q} < 0
\end{aligned}
$$

There is a constant $\epsilon_1$ such that the controller makes the system stable. The proof is completed. □

**Lemma 4 (Schur's lemma) [32].** *Given the appropriate dimensions, we have matrices $\Omega_1$, $\Omega_2$ and $\Omega_3$ also $\Omega_1 = \Omega_1{}^\mathrm{T}$. There are inequalities $\Omega_1 + \Omega_3{}^\mathrm{T}\Omega_2{}^\mathrm{T}\Omega_3 < 0$ that are equivalent to*

$$\begin{bmatrix} \Omega_1 & \Omega_3{}^\mathrm{T} \\ \Omega_3 & -\Omega_2 \end{bmatrix} < 0$$

**Lemma 5 [20].** *For matrix $\Omega \le 0$, both sides of the equation are multiplied by a positive definite matrix $P^{-1}$, that is, $P^{-1}\Omega P^{-1} \le 0$.*

**Theorem 3.** *Combining Lemma 1 and Lemma 2, when there is a positive definite matrix $X > 0$, the uncertain system in nominal mode can achieve robust quadratic stability through optimization and adjustment within the bounded range of parameters. Therefore, the matrix under the condition of uncertain parameters satisfies the following inequality:*

$$\begin{bmatrix} -X + EE^\mathrm{T} & XA^\mathrm{T} & XB^\mathrm{T} & \overline{P}F_a{}^\mathrm{T} & \overline{P}F_b{}^\mathrm{T} \\ AX & -X + EE^\mathrm{T} & 0 & 0 & 0 \\ BX & 0 & -X + EE^\mathrm{T} & 0 & 0 \\ F_a\overline{P} & 0 & 0 & I & 0 \\ F_b\overline{P} & 0 & 0 & 0 & I \end{bmatrix} < 0 \tag{22}$$

**Proof.** The equation of the reasonable positive definite solution is as follows:

$$- P + G^\mathrm{T}PG + H_1{}^\mathrm{T}PH_1 < 0$$

through Lemma 4 and the above proof process, we have the following:

$$- P^{-1} + P^{-1}G^\mathrm{T}PGP^{-1} + P^{-1}H_1{}^\mathrm{T}PH_1P^{-1} < 0 \tag{23}$$

Let $\overline{P} = P^{-1}$, combined with Lemma 3, we have the following:

$$\begin{bmatrix} -\overline{P} & \overline{P}G^{\mathrm{T}} & \overline{P}H_1{}^{\mathrm{T}} \\ G\overline{P} & -\overline{P} & 0 \\ H_1\overline{P} & 0 & -\overline{P} \end{bmatrix} < 0$$

through Equation (12), the above equation becomes:

$$\begin{bmatrix} -\overline{P} & \overline{P}(A+\Delta A)^{\mathrm{T}} & \overline{P}(B+\Delta B)^{\mathrm{T}} \\ (A+\Delta A)\overline{P} & -\overline{P} & 0 \\ (B+\Delta B)\overline{P} & 0 & -\overline{P} \end{bmatrix} < 0$$

take,

$$Z = \begin{bmatrix} -\overline{P} & \overline{P}A^{\mathrm{T}} & \overline{P}B^{\mathrm{T}} \\ A\overline{P} & -\overline{P} & 0 \\ B\overline{P} & 0 & -\overline{P} \end{bmatrix}$$

then,

$$Z + \begin{bmatrix} 0 & \overline{P}\Delta A^{\mathrm{T}} & \overline{P}\Delta B^{\mathrm{T}} \\ \Delta A\overline{P} & 0 & 0 \\ \Delta B\overline{P} & 0 & 0 \end{bmatrix}$$

$$= Z + \begin{bmatrix} 0 & \overline{P}(EMF_a)^{\mathrm{T}} & \overline{P}(EMF_b)^{\mathrm{T}} \\ (EMF_a)\overline{P} & 0 & 0 \\ (EMF_b)\overline{P} & 0 & 0 \end{bmatrix}$$

$$= Z + \begin{bmatrix} EM & & \\ & EM & \\ & & EM \end{bmatrix} \begin{bmatrix} 0 & 0 & 0 \\ F_a\overline{P} & 0 & 0 \\ F_b\overline{P} & 0 & 0 \end{bmatrix} + \begin{bmatrix} 0 & \overline{P}F_a{}^{\mathrm{T}} & \overline{P}F_b{}^{\mathrm{T}} \\ 0 & 0 & 0 \\ 0 & 0 & 0 \end{bmatrix} \begin{bmatrix} (EM)^{\mathrm{T}} & & \\ & (EM)^{\mathrm{T}} & \\ & & (EM)^{\mathrm{T}} \end{bmatrix} < 0 \qquad (24)$$

by Lemma 1, the above inequality applies to the following inequality,

$$Z + \omega \begin{bmatrix} E & & \\ & E & \\ & & E \end{bmatrix} \begin{bmatrix} E^{\mathrm{T}} & & \\ & E^{\mathrm{T}} & \\ & & E^{\mathrm{T}} \end{bmatrix} + \omega^{-1} \begin{bmatrix} 0 & \overline{P}F_a{}^{\mathrm{T}} & \overline{P}F_b{}^{\mathrm{T}} \\ 0 & 0 & 0 \\ 0 & 0 & 0 \end{bmatrix} \begin{bmatrix} 0 & 0 & 0 \\ F_a\overline{P} & 0 & 0 \\ F_b\overline{P} & 0 & 0 \end{bmatrix}$$

$$= \begin{bmatrix} \Omega & \overline{P}A^{\mathrm{T}} & \overline{P}B^{\mathrm{T}} \\ A\overline{P} & -\overline{P} + \omega EE^{\mathrm{T}} & 0 \\ B\overline{P} & 0 & -\overline{P} + \omega EE^{\mathrm{T}} \end{bmatrix} < 0$$

were, $\Omega = -\overline{P} + \omega EE^{\mathrm{T}} + \omega^{-1}\overline{P}F_a{}^{\mathrm{T}}F_a\overline{P} + \overline{P}F_b{}^{\mathrm{T}}F_b\overline{P}$.

Let $\overline{P} = \omega X$, that is

$$\begin{bmatrix} \wedge & XA^{\mathrm{T}} & XB^{\mathrm{T}} \\ AX & -X + EE^{\mathrm{T}} & 0 \\ BX & 0 & -X + EE^{\mathrm{T}} \end{bmatrix} < 0$$

where,

$$\wedge = -X + EE^{\mathrm{T}} + XF_a{}^{\mathrm{T}}F_aX + XF_b{}^{\mathrm{T}}F_bX$$

Using Theorem 2, replace A with $A - BK$ and so on, respectively, and a new LMI similar to (20) appears. The proof of this theorem is completed. The proof is complete. □

**Remark 3.** *The quadratic stabilizability of uncertain systems can be stabilized in the same system, but the opposite is not true.*

Similarly, given constant $\epsilon_1 > 0$, if and only if there is an appropriate scalar function or value, and there are positive definite symmetric matrix $W^{-1}$ and matrix $V^{-1}$, then the guaranteed cost controller is stable to the system within a certain limit.

## 4. Simulation Results

In order to make the model more in line with the standards of the actual engineering field, improve the limitations brought by the nominal model. The uncertain parameters of the filter structure were adjusted in a bounded range to simulate the intermittency of distributed generation, although the controller ensures the stable operation of the system.

First, when two DGs units are connected in parallel and the nominal parameter value is changed, the frequency and voltage will change as shown in the Figure 7:

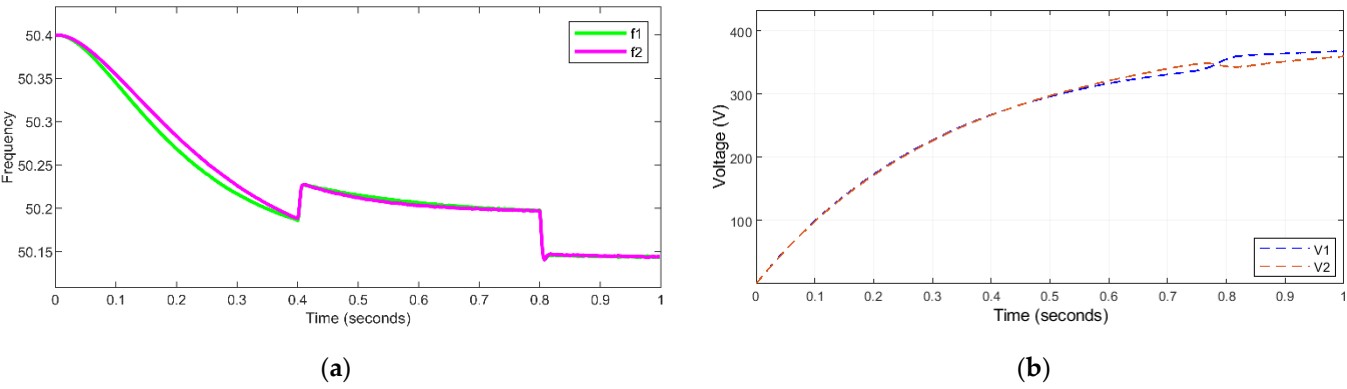

(**a**)                                                                                              (**b**)

**Figure 7.** Generating uncertain parameters: (**a**) frequency; (**b**) voltage.

Within the uncertain and bounded range, adjust the rated values of capacitance and inductance in the filter to ensure that the parameters are between 20% as follow Figure 8 and 50% as follow Figure 9 of the nominal values. The simulation results are as follows.

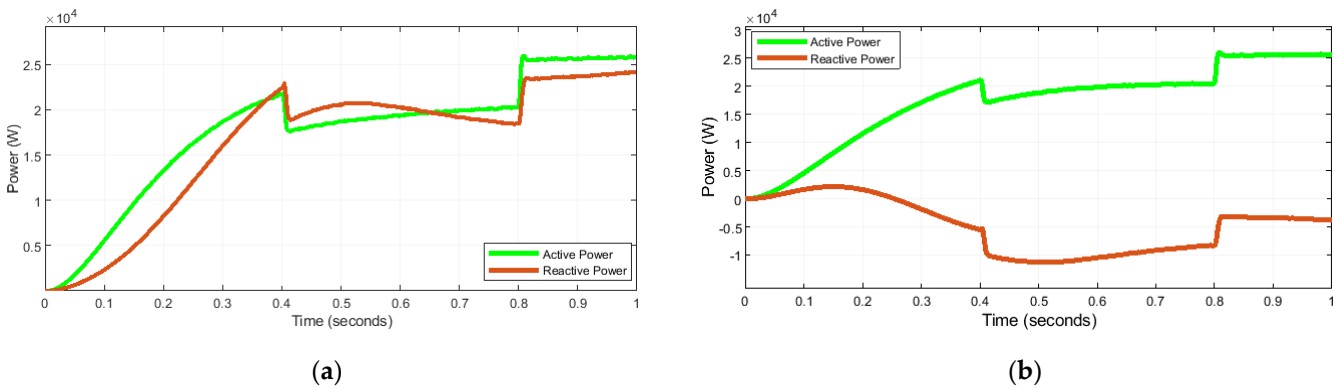

(**a**)                                                                                              (**b**)

**Figure 8.** When the parameters change slightly, the output of active and reactive power: (**a**) DG1; (**b**) DG2.

To make the microgrid system effective and stable, a linear quadratic integral regulator (LQR) controller is designed to regulate the voltage, so that the active and reactive power of the system remain stable within a limited range as follow Figure 10. Through the following experimental simulation analysis, the feasibility and effectiveness of the model are verified.

Within a certain time variation range, for the output power imbalance caused by uncertainty, after the effective control of the controller, the active power and reactive power finally tend to balance, as shown in Figure 11.

Under the same conditions, continuously optimize and adjust the parameter values of the filter structure, and the system can converge more stably, as shown in Figure 12.

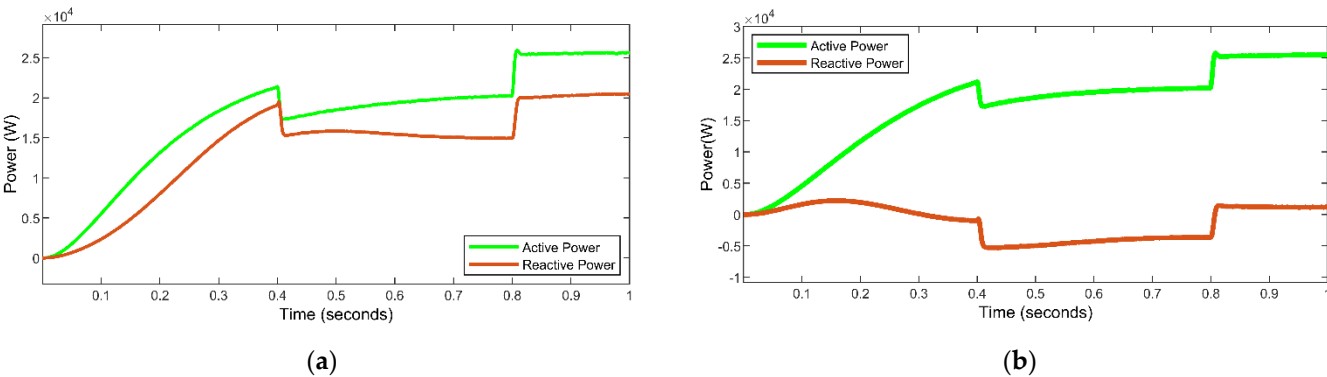

**Figure 9.** When the parameters change greatly, the output of active and reactive power: (**a**) DG1; (**b**) DG2.

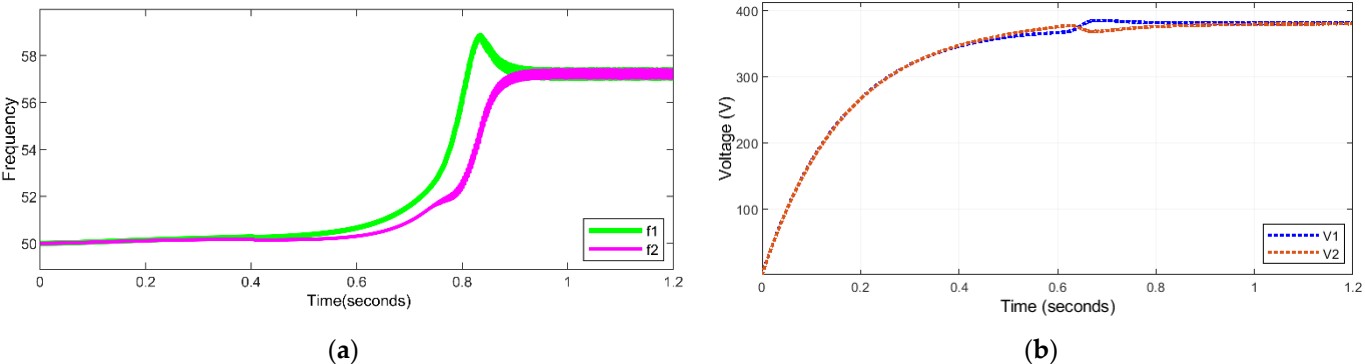

**Figure 10.** The controller adjusts some parameters and performance to optimize and stabilize the system frequency and voltage within a certain range: (**a**) frequency; (**b**) voltage.

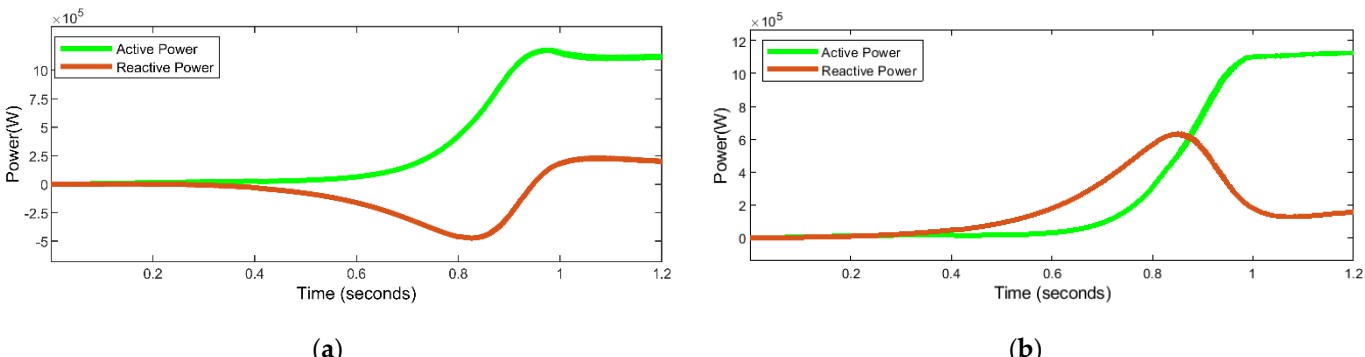

**Figure 11.** The controller adjusts some parameters and performance to optimize and stabilize the active power and reactive power in a certain bounded range: (**a**) DG1; (**b**) DG2.

The performance of capacitors, inductors, and other components in the filter structure changes with temperature. In the case of uncertain parameters, the controller optimizes and adjusts the analog quantity change value within the bounded range of parameters to achieve the purpose of system voltage stability and power balance and to ensure the quality of power output.

At the same time, this method can indirectly and effectively reduce the cost. On the other hand, it could also optimize and adjust the system parameters by the trial-and-error method to determine the boundary value of system stability.

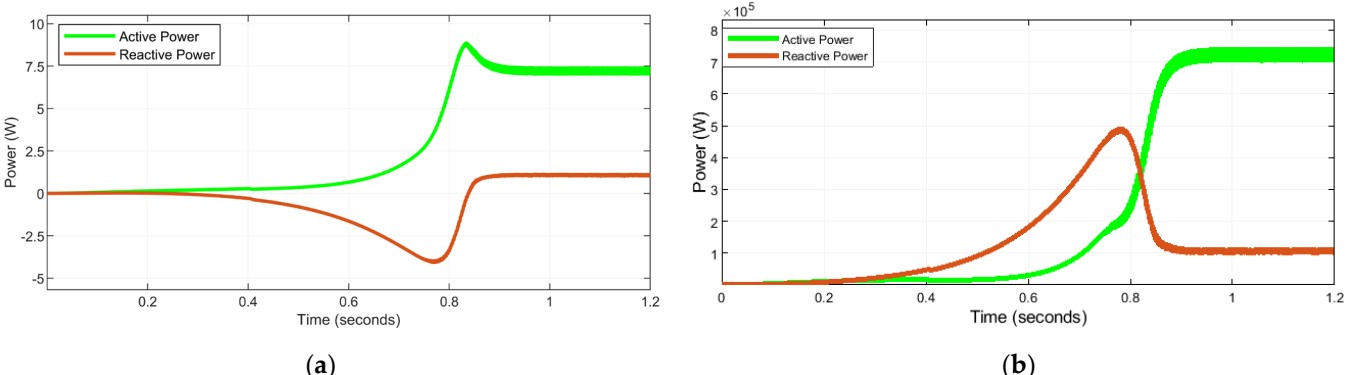

(**a**)　　　　　　　　　　　　　　　　　(**b**)

**Figure 12.** The controller adjusts some parameters and performance to optimize and stabilize the active power and reactive power in a certain bouned range: (**a**) DG1; (**b**) DG2.

## 5. Conclusions

In this paper, the frequency fluctuation of the original system was eliminated through MAF. In the second section, under the condition of nominal parameters, we simulated and verified the stable operation state of voltage and power with one inverter DG unit, and two inverter DG units were run in parallel in the intelligent microgrid system. Secondly, on the basis of the nominal model, the problem of structural parameter uncertainty was introduced. Assuming that the new system was stable and under the initial conditions of guaranteed cost theory, the bounded range of the uncertain parameters was obtained through constraints. Finally, when the system was unstable due to parameter uncertainty, the relevant LQR controller was obtained by the Lyapunov function to give the system a certain progressive stable state. Through simulation, the purpose of optimizing the uncertain model, restraining voltage distortion, and balancing output power was achieved and ensured the stability of the power grid. At the same time, this method could analyze the parameter sensitivity of the controller, continuously adjust the filter parameters, find out the reasonable uncertainty range, verify the hypothesis, effectively simplify the model, and realize the low-cost filter.

To better respond to the needs of the actual project, the follow-up work would consider the impact of load imbalance, faults, communication delay, interference, and other issues on the system. Considering the intermittency of distributed generation, a hardware loop simulation experiment is conducted to simulate the impact of inverter on output power between islanding and grid connected modes. In the hardware in the loop simulation experiment, the impact of the implementation of the switching system on the system performance will also be considered to make the smart microgrid more anti-interference and ensure high-quality and stable output of power.

**Author Contributions:** Conceptualization, Y.F. and Z.M.; methodology, Z.M.; software, Z.M.; validation, Y.F. and Z.M.; formal analysis, Z.M.; resources, Y.F.; data curation, Z.M.; writing—original draft preparation, Z.M.; writing—review and editing, Y.F.; supervision, Y.F.; project administration, Y.F. All authors have read and agreed to the published version of the manuscript.

**Funding:** This study was funded in part by the Key Science and Technology Program of Gansu Province, China. (Grant number 20YF8NA059), in part by the National Natural Science Foundation of China. (Grant number 62141304).

**Data Availability Statement:** Not applicable.

**Acknowledgments:** I want to thank my mentor, Feng, who is easygoing, enthusiastic, rigorous, and careful in his scholarship. In terms of writing the thesis, he always strictly requires me to adhere to "professional standards". From the selection of topics, the determination of topics and objectives, to the repeated modification and polishing of the thesis, Feng has always seriously and responsibly given me profound and detailed guidance, helped me develop research ideas, carefully pointed out errors, and warmly encouraged me.

**Conflicts of Interest:** The authors declare no conflict of interest.

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
