# Peer review of "State-Space Modeling and Analysis for an Inverter-Based Intelligent Microgrid under Parametric Uncertainty"

_applsci, doi:10.3390/app122312418_

Round 1

Reviewer 1 Report (Previous Reviewer 1)

As stated earlier, the authors haven't worked on the suggestions provided. 

1. Keywords are not provided properly and are not sequentially arranged.

2. The major contributions are provided in the introduction. It is recommended to mention the major contributions at the end of the introduction section before the organization of the paper.

3. The literature study is vague. The authors should find the research gap and should present the problem statement and research gap.

4. How can you prove the moving average filter is eliminating the frequency fluctuation?

5. On what basis, the system specifications are selected

6. With the help of the state-space model, the authors can prove the stability of the controller through any frequency response plot

7. An in-depth analysis of simulation experiments is required. I felt the experimental results are not sufficient to claim the novelty

8. What is the future scope of the work proposed?

9. Check for typo errors and Grammatical mistakes that need to be corrected

This is your last chance. If you are not ready to work on the suggestions provided, I cannot recommend the paper for further processing. 

Author Response

Dear  Reviewers,

   Thank you for your warm help!We have read carefully the Reviewers comments in the submission system. Those comments are all valuable and very helpful for revising and improving our paper, as well as the important guiding significance to our researches. We have studied comments carefully and have made correction which we hope meet with approval. Here within enclosed is our careful revision after address each reviewer's comments adequately. Following are our response about reviewer’s comment to our manuscript " State-Space Modeling and Analysis for an Inverter-Based Intelligent Microgrid under Parametric Uncertainty " (ID: applsci-1957513). Revised portion are marked in red in the paper. The main corrections in the paper and the responds to the reviewer’s comments are as following:

Comment 1: Keywords are not provided properly and are not sequentially arranged.

Response 1: We are very sorry for our negligence, the correct keywords are provided and arranged in order.

Comment 2: The major contributions are provided in the introduction. It is recommended to mention the major contributions at the end of the introduction section before the organization of the paper.

Response 2: According to your suggestions, we reorganized the introduction and summarized the main contribution of this paper in the third paragraph.

Comment 3: The literature study is vague. The authors should find the research gap and should present the problem statement and research gap.

Response 3: According to the reviewer's comments. We have revised the introduction and added the contribution of the latest literature related to microgrid. In this part, we state the differences between their research methods and find the advantages and disadvantages of these methods.

Comment 4: How can you prove the moving average filter is eliminating the frequency fluctuation?

Response 4: In order to eliminate the frequency fluctuation of the system, we consider moving average filter to replace the traditional filter. Under the condition that the nominal system operates stably, the natural frequency is stable, as shown in Figure 3 (a).

Comment 5: On what basis, the system specifications are selected.

Response 5: According to the reviewer's suggestion that In order to improve the limitation of the nominal system on practical application, we carried out the design experiment of microgrid parameters uncertain on the basis of this model and reset the parameters of the test system again, as shown in Table 1.

Comment 6: With the help of the state-space model, the authors can prove the stability of the controller through any frequency response plot.

Response 6: According to the your suggestion, Because the frequency parameters in this paper are constant, the changes of voltage and power under the condition of system uncertainty are mainly considered in this paper. As required that designed the controller make the system frequency change as shown in Figure 9 (a). Finally, realized the voltage stability and output power balance of the system.

Comment 7: An in-depth analysis of simulation experiments is required. I felt the experimental results are not sufficient to claim the novelty.

Response 7: According to yours this suggestion. We supplement the results of the simulation experiment, and prove that the method has some reference value.

Comment 8: What is the future scope of the work proposed?

Response 8: According to your suggestions, in the conclusion part of the paper that the second paragraph looks forward to the future work.

Comment 9: Check for typo errors and Grammatical mistakes that need to be corrected.

Response 9: We are very sorry for our incorrect writing that the sentence grammar of the paper is checked and modified. So we were checking and revising the content of the paper as well as possible.

Special thanks to you for your valuable comments.

Reviewer 2 Report (Previous Reviewer 2)

Everything is fine except  english language check.

Author Response

Dear Reviewers,

  Thank you for your warm help! First of all, thank you very much for your affirmation. We carefully checked and corrected the error according to your suggestions and supplemented the article. Please see the attachment!

Round 2

Reviewer 1 Report (Previous Reviewer 1)

The authors have made a reasonable effort to address all the comments raised by the reviewer. I am satisfied with the responses provided. I recommend the paper for further processing. Please check the typo error once again before submitting the final version. 

This manuscript is a resubmission of an earlier submission. The following is a list of the peer review reports and author responses from that submission.

Round 1

Reviewer 1 Report

The following are my comments.

The abstracts look very simple. Rewrite the same by including the contributions, background, and experimental findings.

Keywords are not provided properly and are not sequentially arranged.

The major contributions are provided in the introduction. It is recommended to mention the major contributions at the end of the introduction section before the organization of the paper.

The literature study is vague. The authors should find the research gap and should present the problem statement and research gap.

How can you prove the moving average filter is eliminating the frequency fluctuation?

On what basis, the system specifications are selected

With the help of the state-space model, the authors can prove the stability of the controller through any frequency response plot

An in-depth analysis of simulation experiments is required. I felt the experimental results are not sufficient to claim the novelty

What is the future scope of the work proposed?

Check for the typo errors

Grammatical mistakes need to be corrected

Reviewer 2 Report

The Strengths of this research work are:

-          The problem statement is well-defined.

-          The literature review is good.

-          The references are appropriate.

-          The figures are appropriate.

Weaknesses of this research work are:

-          The introduction must be improved.

-          The related work section must be enhanced.

-          Experimental evaluation must be encouraged.

-          Some improvements are needed in the description of the method.

-          English language is poor

The following points should be incorporated to improve the papers.

·         Some sentences are too long. Generally, it is better to write short sentences with one idea per sentence.

·         The introduction should clearly explain the key limitations of prior work that are relevant to this paper.

·         The authors should explain clearly what the differences are between the prior work and the solution presented in this paper.

·         The authors should first give an overview of their solution before explaining the details.

·         It is necessary to discuss the complexity of the proposed solution.

·         The simulation study should be updated to include some comparison with newer studies.

·         Some text must be added to discuss the future work or research opportunities.